# APE1 Promotes Pancreatic Cancer Proliferation through GFRα1/Src/ERK Axis-Cascade Signaling in Response to GDNF

**DOI:** 10.3390/ijms21103586

**Published:** 2020-05-19

**Authors:** Yoo-Duk Choi, Ji-Yeon Jung, Minwoo Baek, Sheema Khan, Peter I. Song, Sunhyo Ryu, Joo-Yeon Koo, Subhash C. Chauhan, Andrew Tsin, Chan Choi, Won Jae Kim, Mihwa Kim

**Affiliations:** 1Department of Pathology, Chonnam National University Medical School, Gwangju 61186, Korea; drydchoi@chonnam.ac.kr (Y.-D.C.); doctorjennykoo@gmail.com (J.-Y.K.); 2Dental Science Research Institute, Medical Research Center for Biomineralization Disorders, School of Dentistry, Chonnam National University, Gwangju 61186, Korea; jjy@chonnam.ac.kr; 3Department of Pharmacy Practice and Pharmaceutical Sciences, College of Pharmacy, University of Minnesota, Duluth, MN 55812, USA; mbaek@d.umn.edu; 4Department of Immunology & Microbiology, School of Medicine, University of Texas Rio Grande Valley, McAllen, TX 78504, USA; sheema.khan@utrgv.edu (S.K.); subhash.chauhan@utrgv.edu (S.C.C.); 5Department of Molecular Science, School of Medicine, University of Texas Rio Grande Valley, McAllen, TX 78504, USA; peterisong@gmail.com (P.I.S.); andrew.tsin@utrgv.edu (A.T.); 6Division of Pulmonary and Critical Care Medicine, Department of Medicine, Boston University Medical Campus, Boston, MA 02118, USA; sunhyo@bu.edu; 7Department of Pathology, Chonnam National University Hwasun Hospital, Hwasun 58128, Korea; cchoi@chonnam.ac.kr

**Keywords:** APE1, GFRα1, Src/ERK, pancreatic cancer, proliferation

## Abstract

Pancreatic cancer is the worst exocrine gastrointestinal cancer leading to the highest mortality. Recent studies reported that aberrant expression of apurinic/apyrimidinic endodeoxyribonuclease 1 (APE1) is involved in uncontrolled cell growth. However, the molecular mechanism of APE1 biological role remains unrevealed in pancreatic cancer progression. Here, we demonstrate that APE1 accelerates pancreatic cancer cell proliferation through glial cell line-derived neurotrophic factor (GDNF)/glial factor receptor α1 (GFRα1)/Src/ERK axis-cascade signaling. The proliferation of endogenous APE1 expressed-MIA PaCa-2, a human pancreatic carcinoma cell line, was increased by treatment with GDNF, a ligand of GFRα1. Either of downregulated APE1 or GFRα1 expression using small interference RNA (siRNA) inhibited GDNF-induced cancer cell proliferation. The MEK-1 inhibitor PD98059 decreased GDNF-induced MIA PaCa-2 cell proliferation. Src inactivation by either its siRNA or Src inhibitor decreased ERK-phosphorylation in response to GDNF in MIA PaCa-2 cells. Overexpression of GFRα1 in APE1-deficient MIA PaCa-2 cells activated the phosphorylation of Src and ERK. The expression of both APE1 and GFRα1 was gradually increased as progressing pancreatic cancer grades. Our results highlight a critical role for APE1 in GDNF-induced pancreatic cancer cell proliferation through APE1/GFRα1/Src/ERK axis-cascade signaling and provide evidence for future potential therapeutic drug targets for the treatment of pancreatic cancer.

## 1. Introduction

The multifunctional protein apurinic/apyrimidinic endonuclease (APE1) consists of a DNA repair domain at the C-terminus and a redox regulation domain at the N-terminus. The latter functions in the regulation of transcription factors that are involved in cancer promotion and progression such as AP-1, NF-κB, p53, Egr-1, c-Myb, HLF, Pax-8, and STAT3 [1,2,3,4,5]. The level of APE1 protein expression is induced in various cancers including breast cancer, bladder cancer [6], non-small cell lung carcinoma [7,8], glioma [9], pancreatic cancer [1,3,4,10], cervical cancer [11], prostate cancer [12,13], and ovarian cancer [14]. Studies have shown that APE1 plays a role in the cell cycle and in cell survival and it is able to interact with other signaling pathways such as the STAT3 pathway [1,3]. A recent study demonstrated that inhibiting APE1 with E3330, an inhibitor of the APE redox domain, blocks pancreatic tumor growth [15]. Identifying APE1 target molecules is essential for understanding the pathways by which APE1 affects pancreatic cancer progression. Pancreatic cancer is a devastating disease with high mortality among those diagnosed [16]. A lack of early detection screening methods for the disease and its resistance to radiation and chemotherapy contribute to a median 5 year overall survival rate of less than 4% and annual mortality figures that equal the annual rates of incidence in the United States [17].

Glial cell line-derived neurotrophic factor (GDNF) is a member of the neurotrophin polypeptide family. It is secreted by neural tissues and stimulates the development, survival, and differentiation of neuronal cells [18,19]. Glial cell line-derived neurotropic factor (GDNF) exerts its effects on target cells by binding to the glycosylphosphatidylinositol anchor-linked GDNF family receptor α (GFRα) protein which, in turn, recruits the receptor tyrosine kinase RET. In pancreatic cancers, GDNF/GFRα1/RET is expressed more robustly than in normal pancreatic tissue and benign tumors [20,21]. Formation of the GDNF/GFRα1/RET complex triggers RET activation that then stimulates downstream signal transduction pathways, such as the AKT and mitogen-activated protein kinase (MAPK)/extracellular signal-regulated kinase (ERK) pathways; both of these pathways are important for cell invasion, survival, proliferation, and differentiation [20,21,22].

A non-receptor tyrosine kinase Src is a prototype of the Src family of kinases, which includes Src, Lyn, Fyn, Yes, Lck, Blk, and Hck, and it plays an important role in the regulation of proliferation, survival, adhesion, morphology, and motility [23,24,25]. Phosphorylation at Tyr416 in the activation loop of the Src activates its kinase activity, whereas phosphorylation by Csk at Tyr527 in the carboxy-terminal tail renders the enzyme less active [26]. Modulation of MAPK/ERK signaling by Src phosphorylation promotes cell proliferation, migration, and survival [27]. Additionally, knockdown of c-Src expression suppresses pancreatic adenocarcinoma cell proliferation and angiogenesis, although the mechanism by which c-Src regulates pancreatic cancer cell proliferation is still unclear [28]. We demonstrate that APE1 mediates an increase in GFRα1 expression, followed by promoting pancreatic cancer cell proliferation via the Src/ERK signaling pathway.

## 2. Results

### 2.1. APE1 Stimulates GFRα1 Expression to Promote Pancreatic Cancer Cell Proliferation

Previous studies showed that APE1 increased GFRα1 expression by regulating its transcriptional activity. Inhibition of APE1 expression significantly suppressed GFRα1 expression in human pancreatic cancer cell lines, suggesting that APE1 is able to regulate GFRα1 expression in pancreatic cancer cells [4]. However, its mechanism and function in pancreatic cell proliferation are unknown. In this study, we examined whether APE1-mediated regulation of GFRα1 expression has effects on pancreatic cancer cell proliferation using pancreatic cancer cells and human patient tissues, and its mechanism. MIA PaCa-2 human pancreatic cancer cells, which contain endogenous APE1, were treated with GDNF, a binding partner of GFRα1 to investigate the effect of GDNF on pancreatic cancer cell proliferation (Figure 1a). The effect of GDNF was determined as a result of the colorimetric WST-1 cell proliferation assay 24 h after treatment with GDNF (Figure 1a). GDNF promoted cell proliferation in a dose-dependent manner (Figure 1a). In addition, cells were incubated with 50 ng/mL GDNF for the times indicated and cell number was determined using the BrdU assay. The result revealed that GDNF enhanced cell growth in a time-dependent manner and there were noticeable differences in cell growth after 24 h (Figure 1b).

To determine if GDNF-induced cell proliferation requires activation of GFRα1 by APE1, APE1 or GFRα1 expression was silenced in MIA PaCa-2 cells using gene-specific siRNA. At 72 h after GDNF treatment, APE1 siRNA- or GFRα1 siRNA-transfected cells exhibited decreased cell proliferation by approximately 50% compared with control siRNA-transfected cells (Figure 1c,d).

Given that our data showed APE1 can regulate GFRα1, we wished to confirm whether the effect of APE1 on cell proliferation was via GFRα1, therefore GFRα1-encoded lentivirus (MOI, 100) was infected into cells that were transfected with APE1 siRNA. The cells were then treated with 50 ng/mL GDNF for 48 h, and the percentage of cell viability was determined by the WST-1 assay. As shown in Figure 1e,f, overexpression of GFRα1 in APE1-deficient cells rapidly restored cell growth by approximately two-fold as compared with control APE1-deficient cells. These results indicate that GDNF requires GFRα1 expression to promote pancreatic cancer cell proliferation and APE1 can facilitate GDNF/GFRα1-induced cell proliferation through activation of GFRα1.

### 2.2. APE1 Promotes Pancreatic Cancer Cell Proliferation Via a GDNF/GFRα1/ERK Signaling Pathway

Activation of the MAPK/ERK and phosphoinositide 3-kinase (PI3K)/AKT signaling pathways has been implicated in cell proliferation and survival [29,30].

To determine which pathway is involved in APE1-mediated pancreatic cancer cell proliferation, MIA PaCa-2 cells were serum starved for 24 h and incubated with a MAPK/ERK kinase inhibitor (PD98059) or a PI3K inhibitor (Wortmannin) for 30 min prior to treatment with GDNF. As shown in Figure 2a, PD98059, but not Wortmannin, markedly decreased cell proliferation by approximately 50% compared with the GDNF-only treated control. To examine whether ERK is activated by APE1 or GFRα1 in GDNF-treated pancreatic cancer cells, the cells were transfected with APE1 or GFRα1 siRNA for 48 h and then serum starved for 24 h prior to adding GDNF. We observed that treatment with GDNF induced ERK phosphorylation in control siRNA-transfected MIA PaCa-2 cells, but it did not induce ERK phosphorylation in APE1 siRNA- or GFRα1 siRNA-transfected cells (Figure 2b,c). Thus, these results suggest GDNF/GFRα1 activates MAPK/ERK signaling in order to stimulate pancreatic cancer cell proliferation and APE1 can facilitate this major signal transduction pathway through regulation of GFRα1.

### 2.3. APE1-Mediated Pancreatic Cancer Cell Proliferation Requires Activation of Src for Initiation of ERK Signaling

We previously reported that APE1-induced GFRα1 expression increased GDNF-mediated Src phosphorylation in human fibroblast cells [4]. Src kinase regulates MAPK/ERK signaling for cell proliferation [31]. Our data shows that the MAPK/ERK pathway is involved in APE1-mediated pancreatic cancer cell proliferation; therefore, two different methods were used to determine whether Src activation is necessary for APE1-mediated cell proliferation: knockdown of Src expression by siRNA specific for Src and inhibition of Src phosphorylated activation with the chemical inhibitor PP1. MIA PaCa-2 cells were serum-starved for 24 h and then they were either transfected with siRNA for 48 h or incubated with PP1 for 4 h prior to treatment with GDNF. First, Src siRNA was used to evaluate the effect of Src on GDNF/GFRα1-dependent cell proliferation. Scrambled control siRNA had no effect on expression of Src, whereas Src siRNA markedly reduced its expression (Figure 3a). As shown in Figure 3b, Src siRNA knockdown attenuated pancreatic cancer cell proliferation by approximately two-fold (at 48–96 h). Second, MIA PaCa-2 cells were incubated with increasing doses of the Src inhibitor PP1 and the cytotoxicity of PP1 was determined by the WST-1 assay. It did not show cytotoxicity in MIA PaCa-2 cells at a concentration as high as 40 μM (Appendix A). PP1 effectively blocked phosphorylation of Src416 in GDNF-treated cells (Figure 3c), and we found that PP1 attenuated GDNF/GFRα1-stimulated cell proliferation in a concentration-dependent manner (Figure 3d). To determine whether Src kinase’s role in APE1-mediated pancreatic cancer cell proliferation is to regulate ERK signaling, Western blot analysis was performed for phosphorylated ERK in Src siRNA- or PP1-treated MIA PaCa-2 cells in the presence or absence of GDNF. The results showed that both Src and ERK phosphorylation was blocked by either Src siRNA knockdown or PP1 inhibition (Figure 3a,c). To determine the correlation between Src, ERK phosphorylation and APE1, GFRα1 expression, we first downregulated the expression of APE1 or GFRα1 by its siRNA. When we treated APE1-deficient cells with GDNF, we found the level of GFRα1 expression was downregulated (Figure 3e). Simultaneously, phosphorylation of Src and Erk1/2 was markedly reduced in APE1-deficient or GFRα1-deficient cells (Figure 3e–g). In contrast, GFRα1 overexpression restored the phosphorylation of Src and Erk1/2 in APE1-deficient cells (Figure 3e–g). Taken together, these results indicate that Src may be an important intermediary between GDNF/GFRα1 and ERK signaling through APE1-mediated pancreatic cancer cell proliferation.

### 2.4. Coexpression of APE1 and GFRα1 Protein is Elevated in Aggressive Pancreatic Cancer

APE1 has been detected in various malignant cancers including pancreatic adenocarcinoma [32,33]. In addition, GDNF/GFRα1 signaling is detected in pancreatic cancers [21]. To explore the correlation between APE1, GFRα1, and pancreatic cancer progression, immunohistochemical analyses of APE1 and GFRα1 expression were performed utilizing primary tissues of 37 pancreatic adenocarcinoma patients, and its clinicopathological parameters are shown in Table 1 based on different grades and stages. The mean age of these patients was 65.9 years (range 47–77 years). Twelve patients (32.4%) were female, and 25 patients (67.6%) were male. Most tumors were diagnosed in tumor stage pT3 (37 cases, 73%). Five cases (13.5%) were pT2, and the remaining cases were pT1 (four cases, 10.8%) or pT4 (1 case, 2.7%). Most carcinomas obtained from 22 patients (59.5%) were moderately differentiated, while 5 patients (13.5%) were poorly differentiated. 10 patients (27%) were well differentiated. Seventeen patients (45.9%) had no lymph node metastasis (pN0), while 20 patients (54%) had metastases in 1–3 lymph nodes (pN1). As shown in Table 1, all of the 37 cases (100%) were positive for APE1 and GFRα1.

We next investigated the relationship between APE1/GFRα1 expression and clinicopathological data utilizing tissues from pancreatic cancer patients. As shown in Figure 4a,b, the expression of APE1 was weakly detected in nuclei and perinuclear membrane in most normal pancreatic cells. GFRα1 was barely expressed in the nucleus, membrane, and cytoplasm. In contrast, the level of both APE1 and GFRα1 expression was significantly increased in pancreatic adenocarcinoma compared to adjacent normal cells. The enhanced expression of APE1 was detected in nuclei (early to advanced) and cytosol (advanced) of pancreatic adenocarcinoma. As shown in Figure 4c,d, the expression of GFRA1 was increased as progression of stages and grades; whereas, the level of APE1 expression was not dependent on stages. The level of APE1 expression was increased in grade 3 much more than grade 1 and 2. Analysis using the Kaplan–Meier curve showed that the expression pattern of both APE1 and GFRα1 was associated with histological grades and poor survival rate after surgical resection (data not shown). These observations suggest that both of APE1 and GFRα1 are involved in pancreatic cancer progression.

## 3. Discussion

It was previously reported that APE1 increases GFRα1 expression by inducing its transcription through NF-κB signaling in human fibroblast and pancreatic cells [4,34]. We demonstrate that APE1 also stimulates pancreatic cancer cell proliferation via a GDNF/GFRα1/Src/ERK signaling pathway in this study. When APE1 or GFRα1 expression was silenced by siRNA, GDNF-stimulated cell proliferation was attenuated (Figure 1c,d). Intriguingly, GFRα1 overexpression by GFRα1-encoded lentivirus restored GDNF-stimulated cell proliferation in APE-deficient pancreatic cancer cells (Figure 1e,f), indicating that APE1 promotes GDNF-dependent pancreatic cancer cell proliferation by regulating GFRα1 expression.

Ninety percent of pancreatic cancers contain mutations for the Ras small GTPases which regulate Src/ERK signaling, indicating this pathway is crucial to pancreatic cancer development [35,36]. The Src/ERK signaling cascade promotes cell proliferation and migration [27,31]. In this study, we demonstrate that GDNF/GFRα1 trigger cell proliferation via the Src/ERK pathway in pancreatic cancer cells. The proliferative effect of GDNF/GFRα1 was inhibited by PD98059, not Wortmannin (Figure 2a). Moreover, p38 and JNK, but not ERK, were phosphorylated in the presence of GDNF and there was no difference among them (data not shown). Knockdown of Src with siRNA or inhibition of Src activation with PPI inhibitor reduced pancreatic cancer cell proliferation (Figure 3b,d). ERK activation was controlled by Src phosphorylation which promoted cell proliferation (Figure 3a,c). These results suggest that APE1-mediated pancreatic cancer cell proliferation not only needs activation of GFRα1 for the formation of the GDNF/GFRα1 complex, but it also utilizes the Src/ERK cascade.

Although the function of APE1 has been frequently studied in malignant cancers, studies mainly focused on DNA repair and the cell cycle [37]. However, recent reports demonstrated that APE1 regulates STAT3 in pancreatic cancer cell survival using E3330, an APE1 redox inhibitor [1,34,38]. We previously reported that APE1 regulated GFRα1 expression via NF-κB and enhanced pancreatic cell migration and neuronal cell survival [4], and it also participated in JAGGED1/Notch signaling in colon cancer progression by regulating Egr-1 [5]. These reports indicate that APE1 regulates several transcription factors including STAT3, HIF1α, Egr-1, and NF-κΒ and contributes to enhanced cancer progression through various molecular mechanisms by means of its redox domain. It suggests that GFRα1 regulation by APE1 can also be involved in pancreatic cancer progression. We demonstrate in this study that both APE1 and GFRα1 are markedly expressed in adenocarcinoma cells in comparison to normal cells, and their expression seems to be involved in tumor progression (Figure 4).

GDNF signals through a multi-component receptor complex that consists of GFRα and the transmembrane receptor tyrosine kinase (RET) [39]. Four GFRα proteins (GFRα1–4) and four GDNF family growth factors—GDNF, neurturin [NTN], artemin [ART], and persephin [PSP]—have been identified. GFRα1 mainly binds to GDNF, and GFRα2, 3, and 4 bind NTN, ART, and PSP, respectively [39]. Previously, it was reported that the GDNF/GFRα/RET system is involved in tumor cell proliferation, invasion and migration [40,41]. For example, invasion of cancer cells along the nerves is known as perineural invasion (PNI), and it is a common event in some cancers including pancreatic cancer. It has been shown that secretion of GFRα1 by the nerves enhances PNI through the GDNF/RET pathway [40]. However, the role of the GDNF/GFRα/RET system in non-neuronal cells remains unclear. Despite the absence of GFRα1 expression in the normal bile duct, GFRα1 clearly is expressed in bile duct carcinoma, indicating that carcinogenesis leads to the aberrant expression of GFRα1 [41]. In pancreatic cancers, increased levels of neurotrophic factors including GDNF have been described, and these are mainly secreted from intrapancreatic and extrapancreatic nerves [42]. GDNF is strongly expressed in intrapancreatic nerves in the normal pancreas. It is also expressed in pancreatic cancers together with the GDNF receptor RET, and this expression correlates with intrapancreatic neural invasion [21]. In this study, we found that despite GDNF treatment, pancreatic cancer cell proliferation did not occur without GFRα1 expression which is induced by APE1. Overexpression of GFRα1 dramatically restores cellular growth in APE1-deficient pancreatic cancer cells treated with GDNF. Moreover, Src phosphorylation was followed by ERK phosphorylation and acted as a GDNF proliferation mediator. Our study demonstrates that APE1 functions as a modulator of the GDNF/GFRα1/Src/ERK cascade during pancreatic cancer cell proliferation.

## 4. Materials and Methods

### 4.1. Cell Culture

The human pancreatic cancer cell lines BXPC-3 (CRL-1687™, pancreas, adenocarcinoma), PANC-1 (CRL-1469™, pancreas/duct, epithelioid carcinoma), MIA PaCa-2 (CRL-1420™, pancreas, carcinoma), and Capan-2 (HTB-80™, pancreas, adenocarcinoma) were obtained from the American Type Culture Collection (Rockville, MD, USA). All cells were maintained in cell-specific media at 37 °C in a humidified atmosphere of 5% CO_2_.

### 4.2. Reagents

Human recombinant GDNF, PP1, and DMSO were purchased from Sigma (St. Louis, MO, USA). Equivalent DMSO concentrations served as controls. The MAPK/ERK kinase inhibitor PD98059 and the PI3K inhibitor Wortmannin were obtained from Cell Signaling Technology (Beverly, MA, USA). M-PER Mammalian Protein Extraction Reagent were purchased from Pierce (Rockford, IL, USA). The Bio-Rad protein assay was purchased from Bio-Rad (Hercules, CA, USA).

### 4.3. Small Interfering RNA (siRNA)-Based Experiments

The siRNA target sites within the human *APE1* and *GFR*α1 genes were chosen using Ambion’s siRNA target finder program: APE siRNA (534 bp from ATG) 5-GUC UGG UAC GAC UGG AGU Att-3 (sense) and 5-UAC UCC AGU CGU ACC AGA Ctt-3 (antisense); GFRα1 siRNA (1228 bp from ATG) 5-UAC ACA CCU CUG UAU UUC Ctt-3 (sense) and 5-CGU ACG CGG AAU ACU UCG Att-3 (antisense). These siRNAs were prepared using a transcription-based method with a silencer siRNA construction kit (Ambion, Austin, TX, USA). Control siRNA (sc-37007) and c-Src siRNA (sc-29527) were purchased from Santa Cruz Biotechnology as negative controls. The cells were transfected with the siRNA duplexes using Lipofectamine™ RNAiMAX (Invitrogen, Carlsbad, CA, USA).

### 4.4. Lentiviral Construction of GFP or GFRα1

Viral construction was performed as described previously [4]. Briefly, human GFRα1 cDNA was amplified by reverse transcription polymerase chain reaction (RT-PCR) using GFRα1-specific primers: (5′-AAGGAAATAACCACCATGTTCCTGGCGACCCTGTAC-3′ and 5′-TGATGTTTCTGTTAAAGATAATAGGGTGGA-3”) from APE1-transfected GM00637 cells. Green fluorescent protein (GFP) cDNA was amplified by RT-PCR using GFP-specific primers (forward 5′-ATG GTG AGC AAG GGC GAG GAG-3′, reverse 5′-CTT GTA CAG CTC GTC CAT GCC G-3′) from pEGFP-N3 (Clontech, Palo Alto, CA, USA). The GFRα1 and GFP cDNAs were subcloned into pCR8GW/TOPO (K2500-20; Invitrogen) after sequencing. LR recombination reactions using pLenti6/UbC/V5-DEST (V499-10; Invitrogen), viral packaging using 293FT cells, and titration of the full lentiviral vector were performed using the Invitrogen Gateway System and Viral Power Lentiviral Expression System. The presence of GFP and GFRα1 was confirmed by PCR, and correct insertion of the clone was further confirmed by sequencing.

### 4.5. Cell Proliferation Assay

Cells were seeded in 24 well plates at a density of 2 × 10^3^ cells/well. The medium was replaced by DMEM 24 h later and incubated with GDNF (0, 10, 30, and 50 ng/mL) for the indicated times (0, 24, 48, 72, and 96 h) in serum-free medium. In some experiments, cells were preincubated with the Src inhibitor PP1 (10, 20 μM) for 4 h, MAPK/ERK kinase inhibitor PD98059 (10 μM) or the PI3K inhibitor Wortmannin (200 nM) for 30 min before adding GDNF (50 ng/mL) and incubated for the indicated times (0, 24, 48, and 72 h), respectively. The WST-1 cell proliferation reagent was purchased from Roche (Indianapolis, IN, USA). As cells proliferate, more WST-1 is converted to the formazan product. The quantity of formazan dye is directly related to the number of metabolically active cells and can be quantified by measuring the absorbance at Ama× 450 nm in a multiwall plate reader. Additionally, 5-bromo-2′-deoxyuridine (BrdU) uptake by cultured cells was analyzed with BrdU labeling and detection kit from Roche (Basel, Switzerland). Cultured cells were labeled with BrdU at 10 μM for 1 h. The labeled cells were immunostained with anti-BrdU antibody. HRP-labeled anti-mouse antibody was added and the conjugates were visualized with 3, 3′-diaminobenzidine. The number of cells positive for BrdU was counted from five different fields on confocal microscopic photographs (Zeiss, Göttingen, Germany). Experiments were conducted at least in triplicate using separate cultures.

### 4.6. Western Blot Analysis

Equal amounts of total protein were resolved using SDS-PAGE and transferred to nitrocellulose membrane (GE Healthcare Life Sciences, Pittsburgh, PA, USA). The membranes were incubated overnight at 4 °C with primary antibody followed by incubation with a horseradish peroxidase-conjugated secondary antibody. Chemiluminescent detection reagents (GE Healthcare Bio-Sciences, Piscataway, NJ, USA) were used to detect immune-reactive protein. The following antibodies were used: goat anti-hGFRα1 (AF714) from R&D (R&D Systems, Minneapolis, MN, USA), mouse anti-APE1/Ref-1 (c-4), mouse anti-β-actin (C4), mouse anti-c-Src from Santa Cruz Biotechnology (Dallas, TX, USA); mouse anti-phospho-ERK1/2, mouse anti-ERK1/2, rabbit anti-phospho-Src416 from Cell Signaling Technology.

### 4.7. Immunofluorescence

Pancreatic cancer cells were seeded on a slide. The next day, cells were fixed in 4% PFA for 20 min. After being washed with PBS, they were incubated in 0.04% of Triton X-100. After washing with PBS, they were incubated in 0.03% BSA for 10 min at room temperature. Antibodies were applied overnight at 4 °C and washed for 1 h in PBS. Fluorochrome-conjugated or Texas-Red-conjugated secondary antibodies were applied overnight at 4 °C and washed for 1 h in PBS. The slides were rinsed in PBS, mounted in VECTASHIELD (H-1000, Vector Laboratories, Burlingame, CA, USA), and sealed with clear nail varnish. Images were taken by confocal microscopy (Zeiss).

### 4.8. Immunohistochemistry

Paraffin-embedded specimens of tumors and adjacent normal tissues were collected from 37 patients with pancreatic cancer who underwent surgery. The biospecimens and clinical data used for this study were provided by the Biobank of Chonnam National University Hwasun Hospital, a member of the Korea Biobank Network. All samples derived from the Biobank of Chonnam National University were obtained with informed consent under institutional review board-approved protocols. Triplicate core biopsies of 0.6 mm were taken from each donor paraffin block and arrayed. The sections (5 μm thick) were deparaffinized and underwent hematoxylin and eosin staining and immunohistochemistry. After antigen retrieval with 10 mM sodium citrate (pH 6.0), the sections were incubated with mouse polyclonal anti-APE1 and goat anti-GFRα1 antibodies for 24 h at 4 °C. The sections were then incubated with biotinylated secondary antibodies. Antibody labeling was visualized using the ABC kit (Zymed, San Francisco, CA, USA). For each case one complete histological section was evaluated. The percentage of positive cells was scored as: 0 (0%); 2 (<10%); 4 (10%–40%); 6 (41%–60%); 8 (615–80%); 10 (>81%). The staining intensity was scored as: 0 (negative), + (very weak), ++ (weak), +++ (medium), and ++++ (strong). For the immunoreactive score (IRS) the scores for the percentage of positive cells and the staining intensity were multiplicated, resulting in a value between 0 and 10.

### 4.9. Statistical Analysis

The relationship among survival, grade, APE1, and GFRα1 was analyzed by the Kaplan–Meier curve and log-rank test with GraphPad Prism v5.04 (GraphPad Software, La Jolla, CA, USA). Data are presented as mean ± standard deviation. Comparisons among different groups of samples were made using a two-tailed *t*-test, two-ANOVA and the χ2 test. We considered *p* < 0.01 (** & #) and *p* < 0.05 (*) as highly significant.

## 5. Conclusions

We demonstrated a novel regulatory mechanism for GDNF/GFRα1/RET/Src axis by APE1 in pancreatic cancer cell growth. The increased expression of APE1 induced by various cancer risk factors including ROS activates NF-κB1, followed by inducing GFRα1 expression via binding on GFRα1 promoter. GFRα1 and GDNF triggers Ret/Src/ERK activation, resulting in the increased growth of pancreatic cancer cells. Taken together, APE1-induced GFRα1 allows pancreatic cancer cell proliferation through RET/Src/ERK cascade signaling in the response to GDNF. These results highlight a critical role for APE1 in GDNF-induced pancreatic cancer cell proliferation through APE1/GFRα1/Src/ERK axis-cascade signaling and provide evidence for future potential therapeutic drug targets for the treatment of pancreatic cancer (Figure 5).

## Figures and Tables

**Figure 1 ijms-21-03586-f001:**
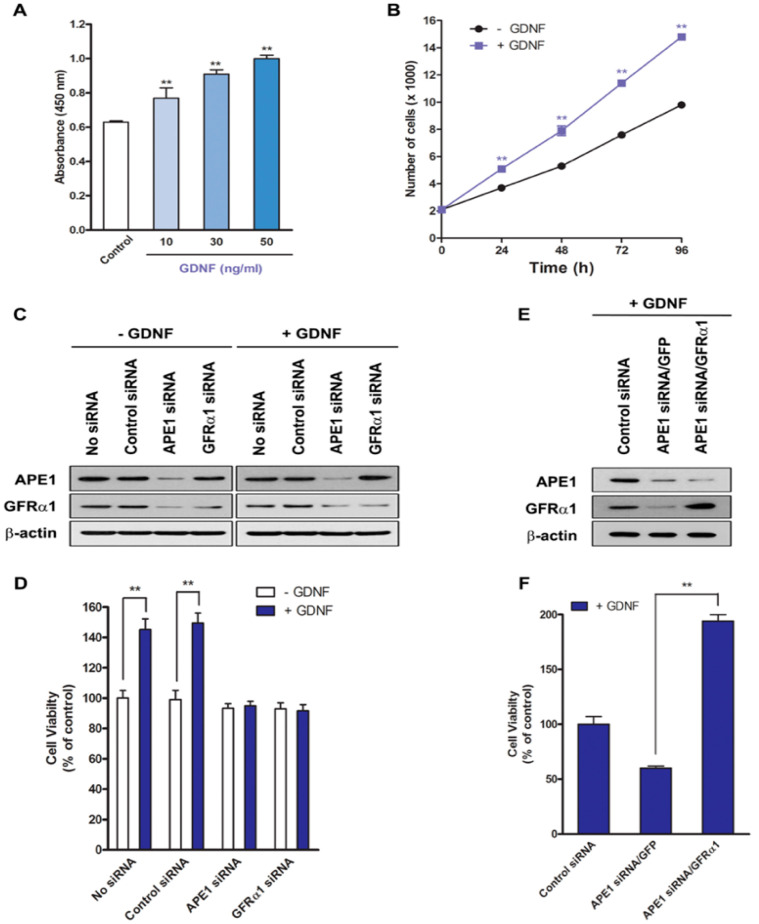
APE1 promotes pancreatic cancer cell proliferation via GFRα1. (**A**) MIA PaCa-2 cells were incubated with GDNF for 24 h. Cell proliferation was analyzed by WST-1 assay. The values are represented as a mean ± standard deviation from three independent experiments. ** Denotes *p* < 0.01 by *t*-test for equality of means. (**B**) MIA PaCa-2 cells were incubated with or without GDNF (50 ng/mL) for up to 96 h. The number of cells was determined by BrdU assay every 24 h after GDNF treatment. The values are represented as a mean ± standard deviation from three independent experiments. ** Denotes *p* < 0.01 by *t*-test for equality of means. (**C** and **D**) MIA PaCa-2 cells were transfected with control siRNA, APE1 siRNA, or GFRα1 siRNA. After transfection, the cells were treated for 48 h with or without 50 ng/mL GDNF. (**C**) Western blot analysis of APE1 and GFRα1. (**D**) Cell proliferation was analyzed by WST-1 assay. The results indicate the percentage of cell proliferation compared with untreated controls (adjusted to 100%) from three independent experiments. ** Denotes *p* < 0.01 by *t*-test for equality of means. (**E** and **F**) Control siRNA (scrambled) or APE1 siRNA-transfected MIA PaCa-2 cells were cultured and transduced with either control GFP or GFRα1 lentivirus. After transfection, the cells were treated for 48 h with or without 50 ng/mL GDNF. (**E**) Western blot analysis of APE1 and GFRα1. (**F**) Cell proliferation was analyzed by WST-1 assay. The results indicate the percentage of cell proliferation compared with untreated controls (adjusted to 100%) from three independent experiments. ** Denotes *p* < 0.01 by *t*-test for equality of means.

**Figure 2 ijms-21-03586-f002:**
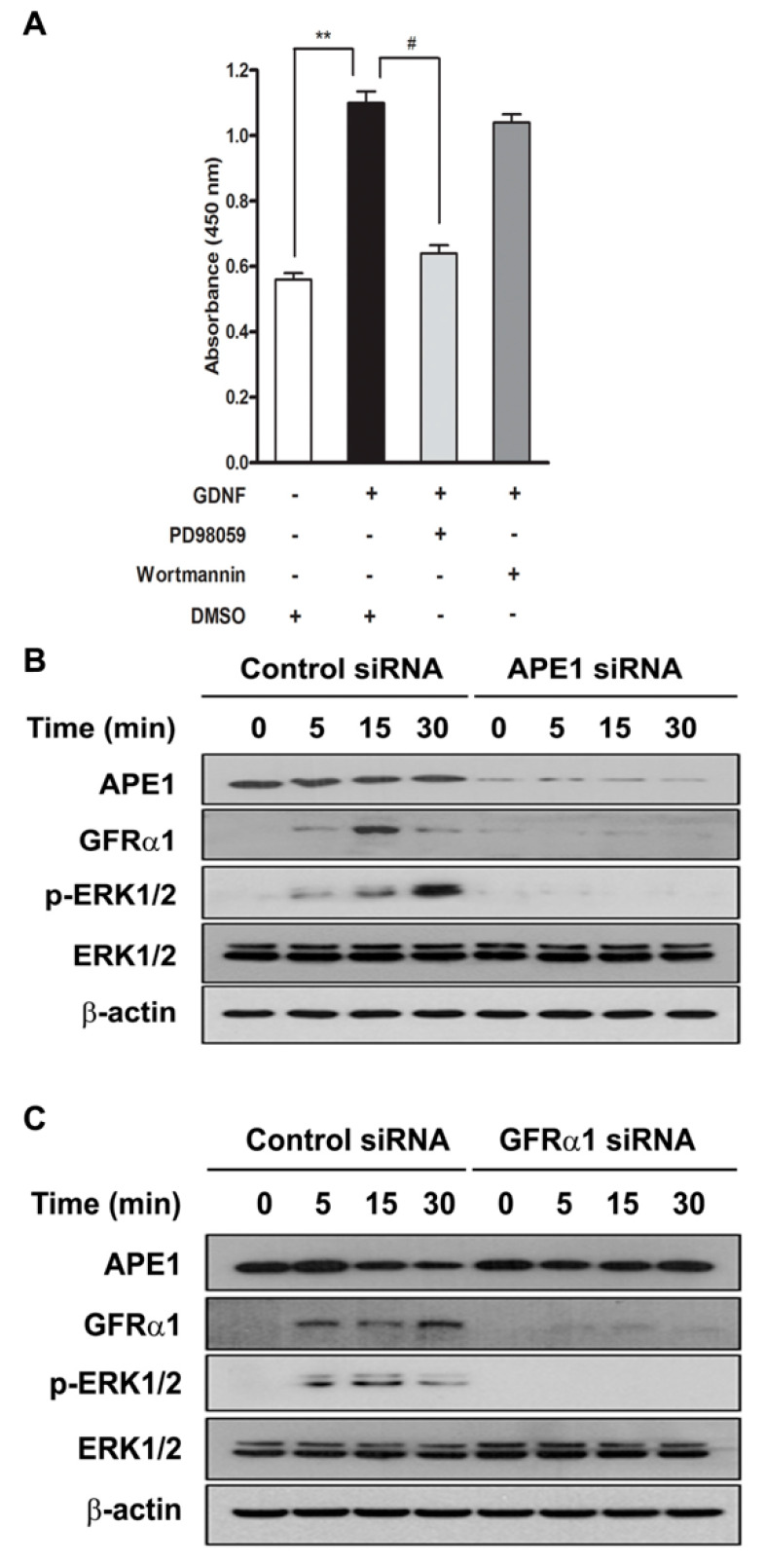
APE1 and GFRα1 increases ERK phosphorylation in response to GDNF. (**A**) MIA PaCa-2 cells were pretreated with MEK-1 inhibitor PD98059 (10 μM) and the PI3K inhibitor Wortmannin (200 μM) and then treated with or without 50 ng/mL GDNF for 24 h. After treatment, cell proliferation was analyzed by WST-1 assay. Data are expressed as mean ± standard deviation from three independent experiments. ** and # denote *p* < 0.01 by *t*-test for equality of means. (**B** and **C**) MIA PaCa-2 cells were transfected with APE1 siRNA (**B**) or GFRα1 siRNA (**C**) and incubated with GDNF (50 ng/mL). Cells were lysed at the indicated times and subjected to immunoblotting with indicated antibodies.

**Figure 3 ijms-21-03586-f003:**
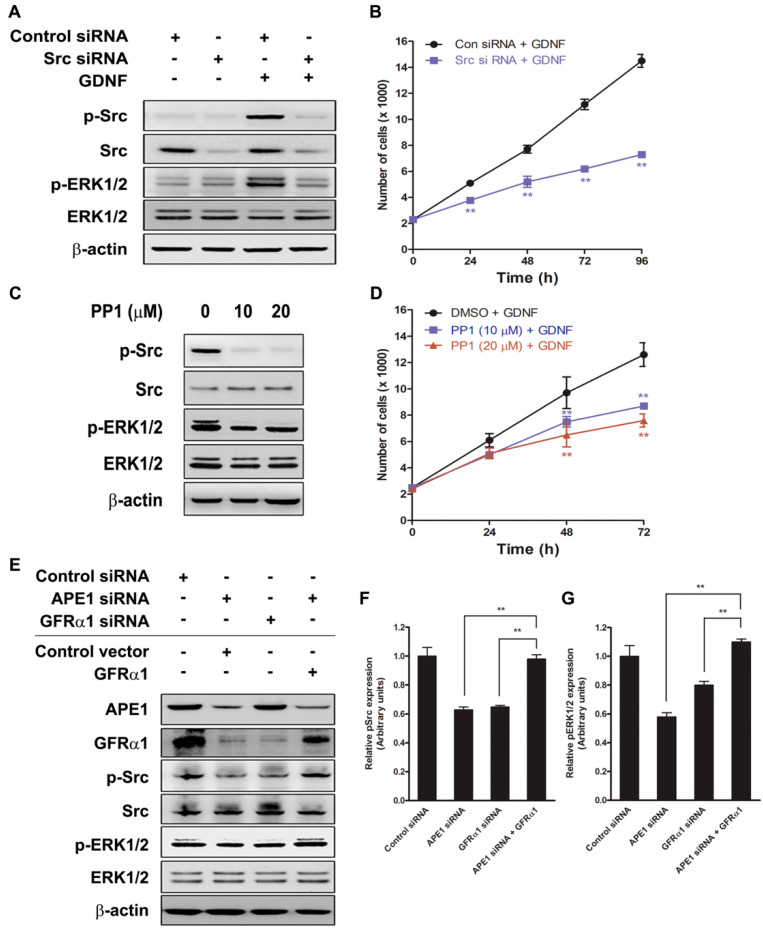
The role of APE1 on Src/ERK phosphorylation via GDNF/GFRα1 signaling. (**A**) MIA PaCa-2 cells were transfected with control siRNA or Src siRNA. Forty-eight hours after transfection, cells were incubated with GDNF (50 ng/mL) for 1 h. Total cell extracts were prepared for immunoblotting with indicated antibodies. (**B**) Control or Src siRNA-transfected MIA PaCa-2 cells were incubated with or without GDNF (50 ng/mL) for up to 96 h. The number of cells was determined by counting the cells every 24 h after GDNF treatment. The values are represented as a mean ± standard deviation from three independent experiments. ** Denotes *p* < 0.01 by *t*-test for equality of means. (**C**) MIA PaCa-2 cells were treated with or without indicated amounts of PP1 and then incubated with GDNF (50 ng/mL) for 1 h. Total cell extracts were prepared for immunoblotting with indicated antibodies. (**D**) MIA PaCa-2 cells were treated with DMSO or PP1 and then incubated with or without GDNF (50 ng/mL) for up to 72 h. The number of cells was determined by BrdU assay every 24 h after GDNF treatment. The values are represented as a mean ± standard deviation from three independent experiments. ** Denotes *p* < 0.01 by *t*-test for equality of means. (**E**–**G**) MIA PaCa-2 cells were transfected with siRNA (control, APE1, or GFRα1), and/or expression vectors (control or GFRα1) and then incubated with GDNF (50 ng/mL) for 1 h. (**E**) Total cell extracts were prepared for immunoblotting with indicated antibodies. (**F**) Relative expression levels of p-Src were quantified by densitometry. The values are represented as a mean ± standard deviation from three independent experiments. ** Denotes *p* < 0.01 by *t*-test for equality of means. (**G**) Relative expression levels of pERK1/2 were quantified by densitometry. The values are represented as a mean ± standard deviation from three independent experiments. ** Denotes *p* < 0.01 by *t*-test for equality of means.

**Figure 4 ijms-21-03586-f004:**
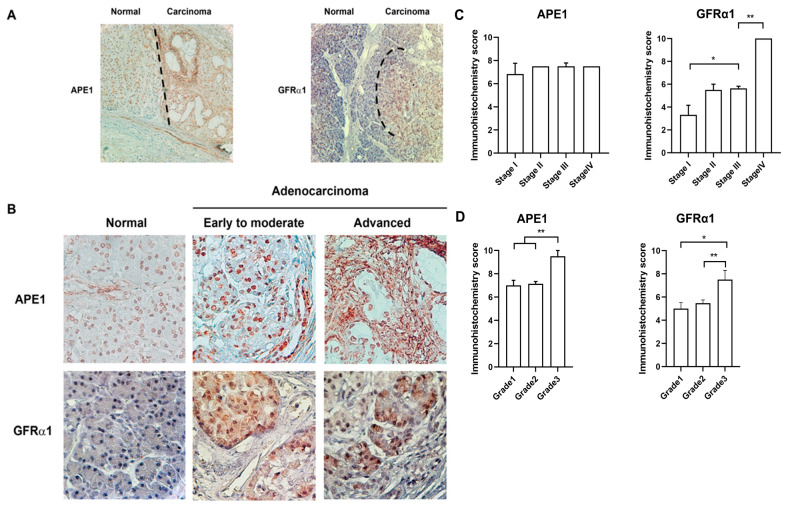
APE1 and GFRα1 expression during pancreatic cancer progression. (**A**) Representative images of APE1 and GFRα1 expression in a normal pancreatic ductal epithelium and pancreatic ductal adenocarcinoma by immunohistochemistry with anti-GFRα1 and anti-APE1 antibodies. Brown staining indicates positive APE1 or GFRα1 staining. Magnification, 10×. (**B**) APE1 and GFRα1 expression in normal pancreatic ductal epithelium, early to moderate and advanced pancreatic ductal adenocarcinoma by immunohistochemistry. (**C**,**D**) The level of APE1 and GFRα1 dependent on TNM stages/grades was assessed by immunohistochemistry scoring. TNM stage system stands for classification of solid tumor using the size and the extension of the spread of primary tumor. (T, size or the direct extension of primary tumor; N, the degree of spread to regional lymph nodes; M, the presence of distant metastasis to other organ beyond regional lymph node). Grades indicate differentiation status (grade 1, well differentiation; grade 2, moderate differentiation; grade 3, poor differentiation). * Denotes *p* < 0.05. ** Denotes *p* < 0.01 by *t*-test and two-way ANOVA for equality of means.

**Figure 5 ijms-21-03586-f005:**
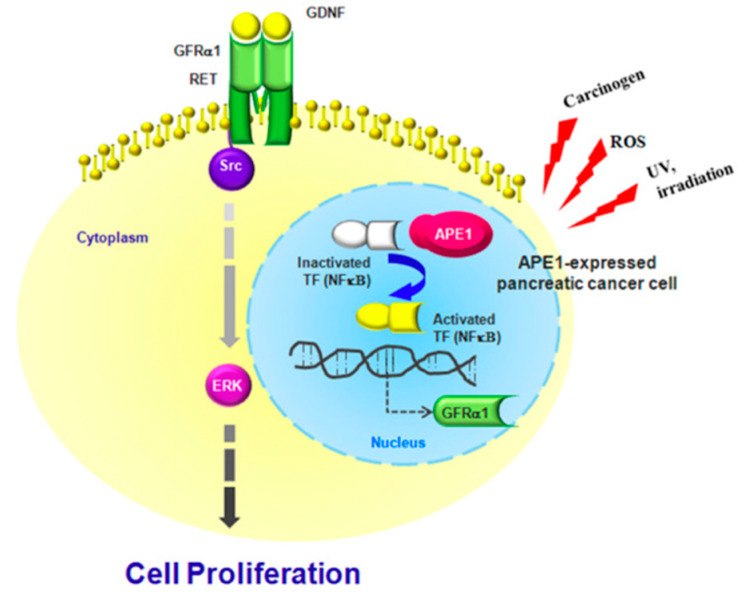
Schematic model of APE1-induced pancreatic cancer cell proliferation. APE1-mediated regulation of GFRα1 expression leads to Src/ERK phosphorylation, which promotes GDNF-dependent pancreatic cancer cell proliferation.

**Table 1 ijms-21-03586-t001:** Relative expressions of APE1 and GFRα1 in pancreatic cancer patients.

Case No.	Gender	Age	Differentiation	TNM Stage	APE1 Expression	GFRα1 Expression
**1**	M	55	G1, Well differentiation	pT_1_N_0_M_x_	++	++
**2**	F	73	G1, Well differentiation	pT_1_N_0_M_x_	++	++
**3**	M	61	G1, Well differentiation	pT_3_N_0_M_x_	++	++
**4**	F	68	G2, Moderate differentiation	pT_3_N_0_M_x_	+++	++
**5**	M	62	G2, Moderate differentiation	pT_3_N_1b_M_x_	+++	+++
**6**	M	49	G2, Moderate differentiation	pT_3_N_1_M_x_	+++	+++
**7**	M	83	G2, Moderate differentiation	pT_2_N_0_M_x_	+++	+++
**8**	M	71	G2, Moderate differentiation	pT_3_N_0_M_x_	+++	+++
**9**	M	64	G2, Moderate differentiation	pT_3_N_1b_M_x_	+++	+++
**10**	M	74	G3, Poor differentiation	pT_4_N_1_M_x_	++++	++++
**11**	F	47	G2, Moderately differentiation	T3N0M0	+++	++
**12**	M	55	G2, Moderately differentiation	T1N0M0	+++	+
**13**	M	66	G2, Moderately differentiation	T2N1M0	+++	++
**14**	F	77	G2, Moderately differentiation	T3N1M0	+++	++
**15**	M	47	G2, Moderately differentiation	T3N1M0	+++	++
**16**	M	76	G2, Moderately differentiation	T3N0M0	+++	++
**17**	M	77	G2, Moderately differentiation	T3N0M0	+++	++
**18**	M	58	G2, Moderately differentiation	T3N1M0	+++	++
**19**	M	72	G2, Moderately differentiation	T3N1M0	+++	++
**20**	F	62	G2, Moderately differentiation	T2N0M0	+++	++
**21**	M	62	G2, Moderately differentiation	T3N0M0	+++	++
**22**	M	71	G1, Well differentiation	T3N1M0	++	+++
**23**	F	71	G1, Well differentiation	T3N1M0	++	+++
**24**	M	67	G3, Poorly differentiated	T3N1M0	++++	+++
**25**	F	58	G1, Well differentiation	T1N0M0	++	+
**26**	M	60	G2, Moderately differentiation	T2N1M0	+++	++
**27**	M	59	G1, Well differentiation	T3N0M1	+++	+
**28**	M	72	G2, Moderately differentiation	T2N1M1	+++	++
**29**	F	67	G2, Moderately differentiation	T3N1M0	++	+
**30**	M	71	G1, Well differentiation	T3N0M0	+++	++
**31**	F	66	G3, Poorly differentiated	T3N1M0	++++	+++
**32**	M	69	G1, Well differentiation	T3N0M0	++	++
**33**	F	63	G1, Well differentiation	T3N1M0	++	++
**34**	M	83	G2, Moderately differentiation	T3N1M0	+++	++
**35**	M	71	G3, Poorly differentiated	T3N1M0	++++	+++
**36**	F	73	G2, Moderately differentiation	T3N1M0	+++	++
**37**	F	60	G3, Poorly differentiated	T3N0M0	++++	++

For each case one complete histological section was evaluated. The staining intensity was scored as: 0 (negative), + (very weak), ++ (weak), +++ (medium), and ++++ (strong).

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
