# Peer review of "APE1 Promotes Pancreatic Cancer Proliferation through GFRα1/Src/ERK Axis-Cascade Signaling in Response to GDNF"

_ijms, 2020, doi:10.3390/ijms21103586_

Round 1

Reviewer 1 Report

  1. In 4.9, the authors considered p < 0.01 (** and #) as highly significant. However, in Fig. 4C and 4D, there're no explanations about *. It would be better that using *, **, *** to show different degree of the statistical difference.  
  2. From Line 172 to Line 175, please elaborate more about the correlation between Src, ERK phosphorylation and APE1, GFRα1 expression, it seems like there's a little bit conflict. 

Author Response

In response to Reviewer #1’s comments, we added the denotation for statistical analysis in line 261-262 and 401-402:

In line 261-262, * denotes p < 0.05, ** denotes p < 0.01 by t-test and Two-way ANOVA for Equality of Means.

In line 401-402, We considered p < 0.01 (** & #) and p < 0.05 (*) as highly significant.

Reviewer 2 Report

The authors demonstrate the protein APE1 in pancreatic cancer proliferation through GDNF/GFRalpha 1 Src/Erk axis.  Though a lot is known in the area but the present manuscript does add an incremental value to our knowledge of the pancreatic cancer progression. The article needs no further improvements.

Author Response

We appreciate for Reviewer 2’s peer review for our new findings of APE1 role in pancreatic cancer proliferation through GDNF/GFRalpha 1 Src/Erk axis.